# Temporary Right-Ventricular Assist Devices: A Systematic Review

**DOI:** 10.3390/jcm11030613

**Published:** 2022-01-26

**Authors:** Mahmoud Abdelshafy, Kadir Caliskan, Goksel Guven, Ahmed Elkoumy, Hagar Elsherbini, Hesham Elzomor, Erhan Tenekecioglu, Sakir Akin, Osama Soliman

**Affiliations:** 1Discipline of Cardiology, Saolta Group, Galway University Hospital, Health Service Executive and CORRIB Core Lab, National University of Ireland Galway (NUIG), H91 V4AY Galway, Ireland; mahmoud.abdelshafy@nuigalway.ie (M.A.); Ahmed.Elkoumy@nuigalway.ie (A.E.); hesham.elzomor@nuigalway.ie (H.E.); 2Department of Cardiology, Al-Azhar University, Al-Hussein University Hospital, Cairo 11311, Egypt; 3Department of Cardiology, Erasmus MC University Medical Center, 3015 GD Rotterdam, The Netherlands; k.caliskan@erasmusmc.nl (K.C.); hagaar.x@hotmail.com (H.E.); drercardio2@gmail.com (E.T.); sakirakin@gmail.com (S.A.); 4Department of Intensive Care Medicine, Erasmus MC University Medical Center, 3015 GD Rotterdam, The Netherlands; drgoksel@hotmail.com; 5Division of Internal Medicine Intensive Care, Hacettepe University Faculty of Medicine, 06230 Ankara, Turkey; 6Islamic Center of Cardiology and Cardiac Surgery, Al-Azhar University, Nasr City, Cairo 11651, Egypt; 7Department of Cardiology, Bursa Education and Research Hospital, Bursa Medicine School, University of Health Sciences, 16059 Bursa, Turkey; 8Department of Intensive Care, Haga Teaching Hospital, 2545 AA The Hague, The Netherlands; 9CÚRAM, The SFI Research Centre for Medical Devices, H91TK 33 Galway, Ireland

**Keywords:** right-sided heart failure, temporary right ventricular assist device, safety, efficacy, systematic review

## Abstract

Acute right-sided heart failure (RHF) is a complex clinical syndrome, with a wide range of clinical presentations, associated with increased mortality and morbidity, but about which there is a scarcity of evidence-based literature. A temporary right-ventricular assist device (t-RVAD) is a potential treatment option for selected patients with severe right-ventricular dysfunction as a bridge-to-recovery or as a permanent solution. We sought to conduct a systematic review to determine the safety and efficacy of t-RVAD implantation. Thirty-one studies met the inclusion criteria, from which data were extracted. Successful t-RVAD weaning ranged between 23% and 100%. Moreover, 30-day survival post-temporary RAVD implantation ranged from 46% to 100%. Bleeding, acute kidney injury, stroke, and device malfunction were the most commonly reported complications. Notwithstanding this, t-RVAD is a lifesaving option for patients with severe RHF, but the evidence stems from small non-randomized heterogeneous studies utilizing a variety of devices. Both the etiology of RHF and time of intervention might play a major role in determining the t-RVAD outcome. Standardized endpoints definitions, design and methodology for t-RVAD trials is needed. Furthermore, efforts should continue in improving the technology as well as improving the timely provision of a t-RVAD.

## 1. Introduction

Increasing use of advanced heart failure (HF) treatments including temporary and durable mechanical circularity support (MCS) are mostly directed to left-sided HF. However, concomitant or primary right-sided heart failure (RHF) are not uncommon and increases significantly morbidity and mortality [1]. This is especially vitally important in the era of long-term, durable left-ventricular assist devices (LVAD) solutions. RHF prevalence and associated mortality vary according to the underlying cause. Common clinical phenotypes are acute-on-chronic right-ventricular dysfunction with or without left-ventricular cardiomyopathy, myocardial infarction or ischemia, post-cardiotomy (PCCS), post LVAD implantation, post myocarditis, post-heart transplantation (HTX), and pulmonary thromboembolic events [1,2,3]. Management of RHF is usually medical, but in patients with imminent or full-blown cardiogenic shock, mechanical circulatory support including a right-ventricular assist device (RVAD) is indispensable. RHF usually requires temporary support in contrast to left-ventricular failure as its main pathophysiologic mechanisms, adaptive remodeling and impaired contractility, could be reversible along with lacking of established long-term, durable RVAD solutions [4]. RVAD could be surgically or percutaneously implanted and according to the duration of support could be short-term (temporary) or (rarely) long-term [1]. Despite several clinical RVAD devices, the existing literature is limited with rarely small prospective randomized trials, heterogenous, and conflicting studies, and single center cohorts. In this systemic review, we aimed to elucidate the contemporary literature for the safety and efficacy of using a temporary right-ventricular assist device (t-RAVD) in different clinical scenarios and RHF phenotypes.

## 2. Materials and Methods

### 2.1. Search Strategy

This systematic review was performed and reported according to the Preferred Reporting Items for Systematic Reviews and Meta-Analyses (PRISMA) guidelines [5]. From inception to November 2021 we searched Embase, Medline Ovid, Web of Science, Cochrane CENTRAL register of trials, for articles published until the date of search. The full search methodology is available in Appendix B.

### 2.2. Inclusion and Exclusion Criteria

Adult patients ≥18 years (except one paper [6] with three patients of whom one was 14 years and two 16 years old) with RHF supported with a t-RAVD were included. We included all clinical studies (e.g., randomized trials, observational cohorts, case-control, case-series) containing ≥10 patients and published from 2005 and beyond, which marks the introduction of continuous-flow left ventricular assist devices (LVAD) thereby changing the landscape of severe HF and cardiogenic shock treatment. Case reports, editorials, reviews, and articles that were not in the English language were excluded. In the case of multiple publications on the same patient cohort, the most recent publication was included in the analysis.

### 2.3. Study Selection

The search results were transferred for Rayyan—a web and mobile app for systematic reviews to remove duplications and perform the screening [7]. Two researchers (M.A. and G.G.) independently reviewed abstracts and full texts in a blinded standardized manner. Disagreements between the researchers to include a study were discussed and resolved before final approval. Furthermore, references in selected articles were independently crosschecked by the two researchers for other relevant studies.

### 2.4. Data Extraction

Two authors (M.A. and A.E.) extracted the data independently, using a pre-defined standardized data extraction form. These extracted data were compared and confirmed with the original articles. Data extraction included study characteristics (e.g., author and study design), number of patients, and type of t-RVAD, and duration of t-RVAD. Furthermore, patient demographics (e.g., age and sex), survival outcomes (survival to discharge, survival to weaning, 30-day survival, and 180-day survival). Likewise, successful t-RVAD weaning, switch to permanent RVAD and successful bridging to heart transplantation (HTx) were documented. The following endpoints were extracted if available: length of ICU stay, end-organ dysfunction, bleeding, stroke, device malfunction, infection, hemolysis, pulmonary embolism, or hemorrhage, new moderate or more tricuspid regurgitation, and arrhythmia were collected when available. Microsoft Office Excel was used for data extraction. Due to the significant heterogeneity between studies, the pooling of data for meta-analysis was not deemed appropriate.

## 3. Results

The search strategy resulted in 11,545 studies. After the removal of duplicates, 6429 studies remained. After reviewing the title and abstract, another 6365 studies were removed due to irrelevance. Of the remaining 66 studies, 31 studies [6,8,9,10,11,12,13,14,15,16,17,18,19,20,21,22,23,24,25,26,27,28,29,30,31,32,33,34,35,36,37] met the predefined inclusion criteria and were consequently included in this review. Figure 1 displays the PRISMA flowchart. Metadata of the 31 studies included in the systematic review are shown in Appendix A.

### 3.1. Patients’ Characteristics

The patient characteristics in the selected studies including the number of patients in each study, sex, age, t-RVAD indications, underlying disease of patients post LAVD, and the follow-up duration are listed in Table 1.

### 3.2. Types of Temporary Right-Ventricular Assist Device (t-RVAD) and Approach to t-RVAD Implantation

ProtekDuo (TandemLife, Pittsburg, PA, USA), Impella RP (Abiomed, Danvers, MA, USA), TandemHeart (TandemLife, Pittsburg, PA, USA), and CentriMag (Levitronix LLC, Waltham, MA, USA) were the commonly reported t-RVAD devices in literature. Table 2 lists the key design characteristics, hemodynamic performance of the clinically used t-RVADs.

#### 3.2.1. ProtekDuo

Percutaneous implantation of ProtekDuo RVAD was reported in seven studies including 146 patients [8,10,11,12,14,17,18]. ProtekDuo is a 29 F or 31 F, dual-lumen cannula that is inserted percutaneously through the right internal jugular vein under fluoroscopic and ultrasound guidance. The distal end of the cannula is positioned in the main pulmonary artery over a stiff guidewire. The inflow ports are positioned in the right atrium and the distal outflow ports is positioned in the main pulmonary artery. The procedure can take place in the cardiac catheterization laboratory or in the operating room. ProtekDuo can be connected to a variety of extracorporeal devices, creating a t-RVAD with or without an oxygenator [8,10,11,12,14,17,18].

#### 3.2.2. Impella RP

Percutaneous implantation of Impella RP was reported in seven studies including 110 patients [6,9,17,20,29]. The Impella RP is a 22 F 3-dimensional catheter-based microaxial pump mounted on an 11 F catheter. The catheter pump is advanced antegrade under fluoroscopic guidance over a 0.025-inch platinum wire with the aid of a flow directed catheter and positioned across the tricuspid and pulmonary valves through a 23 F peel-away sheath inserted in the femoral vein [6,9,17,20,29].

#### 3.2.3. Impella RD

One study [29] reported using Impella RD in a cohort of 15 patients. The Impella Recover RD is a microaxial pump designed specifically for temporary RV support. It is surgically implanted through a sternotomy, with direct anastomosis of the inflow cannula (41 F) to the right atrium, and outflow cannula (24 F) to the pulmonary artery. The pump is connected to a mobile console by a 9 F catheter containing the driveline and purge line. The Impella RD is no longer available for clinical use [29].

#### 3.2.4. TandemHeart RVAD (TH-RVAD)

The TH-RVAD is an extracorporeal centrifugal-flow pump. Seven studies [8,10,11,12,14,17,18] reported using TH-RVAD with ProtekDuo. One study [34] reported percutaneous implantation of TandemHeart without ProtekDuo, which was done with two venous cannulas to deliver blood from the RA to the main PA. Most TH-RVAD cannulas are deployed via both femoral veins. For bilateral femoral cannulation, the outflow cannula is placed in the main PA via the right femoral vein, and the inflow cannula is placed in the RA via the left femoral vein. In patients with long torsos, (distance from the femoral vein to the fifth intercostal space exceeds 58 cm), the outflow cannula can be placed in the main PA via the right internal jugular vein. This approach can also be used if limitations to femoral venous access exist, including infection, thrombosis, or inferior vena caval filters [3].

#### 3.2.5. Surgically Implanted t-RVADs

One study [36] reported using Capiox or Gyropump. One study [33] reported using Rotaflow. The remaining studies reported using CentriMag only or with other types of devices e.g., Deltastream, AB5000, Thoratec PVAD and Rotaflow.

Surgical implantation of t-RVAD was reported either via direct cannulation of the right atrium (RA) and pulmonary artery (PA) or by peripheral cannulation of the femoral vein [25,26,35] and a percutaneous pulmonary arterial cannula (remote cannula removal approach) or direct cannulation of the PA. One study [22] reported using minimally invasive surgical approach via left-sided mini-thoracotomy to suture the outflow cannula of t-RVAD with PA and t-RAVD inflow via femoral vein. Two studies [13,23] did not report details about the approach of t-RVAD implantation.

#### 3.2.6. Timing of sRVAD Implantation

The timing of sRVADs was reported in the studies included in our review in different manners: intra-operative pre-planned surgical implantation was reported in two studies [16,31]. Bailout intra-operative (concurrent/simultaneous) t-RVAD implantation was reported in seven studies based on the development of RHF according to the local institute protocol [17,22,25,26,32,33,36]. Both intra-operative pre-planned and bail-out (based on the development of RHF) sRVAD implantation was reported in one study [27]. Two studies [15,21] compared intra-operative (concurrent/simultaneous) vs. staged t-RAVD (within 7 days after LVAD insertion) [15] or just post-operative insertion of t-RVAD [21]. One study [19] compared post-operative (immediate): within 24h of being indicated for RVAD while post-operative (delayed). Both intra-operative (concurrent/simultaneous) and post-operative sRVAD implantation was according to the clinical situation and the local criteria of RHF diagnosis [28,34].

#### 3.2.7. Oxy t-RVAD

Four studies [24,25,26,35] including 61 patients reported using of oxygenator in the t-RVAD circuit. One study [24] compared using of t-RVAD with or without oxygenator.

### 3.3. Indication for t-RVAD Implantation (Patient’s Phenotypes)

Seventeen studies [10,13,14,15,17,21,22,23,24,25,27,28,30,31,32,33,36] reported indications for implantation of t-RVAD due to post-LVAD RHF only. One study [11] reported using t-RVAD post myocardial infarction. One study [16] reported prophylactic t-RVAD insertion during primary operation for high-risk valve disease. The remaining studies [6,8,9,12,18,19,20,26,29,34,35,37] reported variable indications of t-RVAD insertion among mixed RHF patient phenotypes (Table 1).

### 3.4. Duration of t-RVAD Support

The duration of t-RVAD support was reported in 25/31 studies in this review and it was reported as either mean ± SD or median and it was ranged from 5 h to 400 days (Table 3).

### 3.5. Survival Endpoints

Reporting of survival rate widely varied among different studies (Table 4). Survival rate at 30-day ranged between 46% and 100% in 16 studies [8,9,11,13,14,15,16,17,19,20,22,23,24,27,29,31] Survival to weaning was reported in 9 studies [12,16,21,22,25,26,31,35,37] ranging between 40% and 100 %. Survival to discharge was reported in 11 studies [6,12,16,21,25,26,28,32,33,34,35] ranging between 38% and 100%. Furthermore, 10 studies [8,13,16,20,21,24,27,30,31,36] reported 180-day survival ranged between 50% to 92%. The death at end of follow up period was reported in 16 studies [6,8,10,11,14,15,16,18,19,20,21,22,25,26,32,33] ranging between 20% to 52%.

### 3.6. Outcome Other Than Survival

Successful t-RVAD weaning (Table 5) ranged between 23% and 100% in 21 studies [6,8,10,11,12,14,15,16,18,19,21,22,24,25,28,29,31,32,35,36,37]. Furthermore, switch to permanent RVAD was reported in five studies [10,11,15,18,31] ranging between 7% and 35%. Four studies [15,26,30,31] reported the rate of successful bridging to heart transplantation ranging between 9 % to 69%.

### 3.7. Complications following t-RVAD Implantation

Acute kidney injury, post-operative bleeding, stroke and device malfunction were the most commonly reported complications. The main complications reported in all studies are summarized in Table 6.

#### 3.7.1. End Organ Dysfunction Post t-RVAD Implantation

Fourteen studies [8,10,11,15,16,17,21,22,25,26,28,30,31,33] reported the incidence of acute kidney injury or renal replacement therapy as a complication post t-RVAD implantation ranging between 0% to 80%. One study [26] reported 10% for the incidence of refractory hepatic failure.

#### 3.7.2. Bleeding Post t-RVAD Implantation

Different definitions and forms of post t-RVAD implantation bleeding was reported in different studies including major bleeding in 11 studies [11,12,13,15,20,22,25,26,31,34,35] ranging between 0% to 48%; gastrointestinal bleeding in five studies [8,16,18,22,25] ranging between 0% to 46% and reoperation for bleeding in six studies [11,16,19,25,28,33] ranging between 0% to 40%.

#### 3.7.3. Device Malfunction Post t-RVAD Implantation

Five studies [6,10,12,15,22] reported the incidence of device thrombosis post t-RVAD implantation ranging between 0% and 16%. Three studies [10,12,29] reported the incidence of cannula migration requiring reposition as 7.5%, 7% and 5.6% respectively. One study [13] reported 97% as freedom from device malfunction at one month. However, it is not mentioned if device malfunction was exclusive for t-RVADs or associated with LVADs malfunction.

#### 3.7.4. Neurological Outcome Post t-RVAD Implantation

Ten studies [8,13,15,19,21,25,26,33,35,38] reported the incidence of stroke post-t-RVAD implantation ranging between 0% and 23%. Six studies [11,14,18,22,25,31] reported specifically the incidence of haemorrhagic stroke post t-RVAD implantation ranging between 2% and 12%.

#### 3.7.5. Sepsis Post-t-RAVD Implantation

Ten studies [8,13,15,19,21,25,26,31,33,35] reported the incidence of sepsis post-t-RVAD implantation ranging between 0% to 64%. Moreover, two studies [11,25] reported the incidence of chest infection as 40% and 20%, respectively. One study [15] reported 19% as the incidence of driveline infection requiring readmission to the hospital in the whole study cohort.

#### 3.7.6. Other Reported Complications

Five studies [6,9,10,20,29] reported the incidence of haemolysis ranging between 15% and 42%. Additionally, five studies [15,20,25,26,31] reported the incidence of pulmonary haemorrhage ranging between 0% and 20%. Superior vena cava syndrome incidence reported in two studies [10,12] as 0% and 7.5% respectively. The incidence of more than or equal new moderate tricuspid regurgitation reported in two studies [10,20] as 36% and 2.9%. Moreover, three studies [19,21,27] reported the incidence of arrythmia as 45%, 37 and 6%, respectively.

## 4. Discussion

To the best of our knowledge, this systematic review is the first to assess the outcome of different t-RVADs in different indications and patient phenotypes. the key findings are:1.We included 31 studies comprising 1598 patients in this systematic review2.Successful t-RVAD weaning was reported in our review between 23% and 100%. Moreover, 30-day survival post temporary RAVD implantation ranged from 46% to 100%.3.Evidence stems from non-randomized heterogeneous trials which makes the comparison between the different studies or pooling the data in meta-analysis is not possible.4.Acute kidney injury, post-operative bleeding, stroke, and device malfunction were the most commonly reported complications.5.Subgroup analyses are obviously not adequately powered to investigate the determinants of device success.

### 4.1. Efficacy of t-RVAD in This Systematic Review

Survival rate following t-RVAD implantation varied across different studies. It is likely related to the primary cause of RHF, the severity of end-organ dysfunction, and much less dependent on adverse events and complications related to t-RVAD. In addition, the timing of t-RVAD implantation probably plays a major role in the survival post t-RVAD. Planned t-RVAD implantation in patients at high risk of RHF likely remains the optimal strategy to improve survival. Similarly, even when t-RVAD implantation is unplanned, early t-RVAD implantation in patients with acute severe RHF are likely to improve the chance for survival than delayed implantation. Percutaneous novel developed t-RVAD represent a good option in those patients in the perioperative period [17].

In our systemic review, the largest data on t-RVAD was derived from the EUROMACS and the INTERMACS registries. Vierecke et al. [13] reported a 30-day survival of 73% in 342 patients with LVAD and t-RVAD in the EUROMACS registry. Kiernan et al. [23] reported a 30-day survival of 73.5% in 386 patients with LVAD and t-RVAD in the INTERMACS registry. A smaller study reported extreme rates of survival including Jaidka et al. [16] who reported 100% 30-day survival in 10 patients at high risk for RHF (severe baseline right ventricular dysfunction or systemic pulmonary artery pressures) underwent valve surgery and prophylactic CentriMag insertion. Likewise, the lowest 30-day survival of 45.5% was reported by Schaefer et al. [22] in 11 LVAD patients treated with t-RVAD via sternotomy in the comparative arm of the study while the 30-day survival was 80% in the main group of 10 LVAD patients treated with minimally invasive t-RVAD.

Data on the comparison of percutaneous and surgical t-RVAD is scarce. Coromilas et al. [17] reported 84.2% 30-day in 19 LVAD patients with t-RVAD percutaneously inserted mixed group of ProtekDuo (*n* = 15) and Impella RP (*n* = 4) versus 66.7% 30-day survival in 21 LVAD patients with CentriMag surgically inserted.

Likewise, data on the optimal timing of t-RVAD insertion is scarce. Khorsandi et al. [15] reported higher 30-Day survival (93.1% vs. 71.4%) in patients who received t-RVAD as concurrent (with LVAD surgery) vs. delayed (staged after LVAD) t-RVAD insertion. Moreover, Bhama et al. [19] reported 90-day survival was better (79% vs. 46%) for patients who received immediate (within 24 h of being indicated to RVAD) in comparison to delayed (after 24 h of being indicated to RVAD) t-RVAD support. Takeda et al. [28] reported numerically higher but statistically insignificant successful t-RVAD weaning (54% vs. 38% *p* = 0.3) for concurrent vs. delayed t-RVAD insertion. The timing of a temporary RVAD insertion is probably of utmost importance. The RVAD should be inserted as early as possible before the development of secondary organ/multi-organ failure. The best strategy depends on a good prediction of RHF, which is not established yet. One step further, the t-RVAD implantation should be probably planned prophylactic in intermediate to high-risk patients, including the selection of the best appropriate device and the best technique. In fact, the RVAD device and implantation technique selection, percutaneous versus surgical depend on the situation and the availability of the devices with consideration of the advantages offered by the percutaneous RVAD as they are less invasive, easy insertion and easy removal particularly if the RHF occurs in the ICU or perioperatively [38].

Another important aspect that might affect the t-RVAD outcome is RHF patient’s phenotypes. Overall, post-LVAD RHF patients have higher 30-day survival than other phenotypes [9,19,20,29] and higher survival to weaning in one study [37]. However, this subgroup analysis is not adequately powered to detect meaningful difference. In contrast, one study [9] reported a higher (80% vs. 67%) 30-day survival in patients with non-ischemic cardiomyopathy. Likewise, one study [26] reported a higher (75% vs. 47%) one-year survival in the post-cardiotomy patients (*n* = 4) than patients requiring t-RVAD support following LVAD implantation (*n* = 17). Although some studies [13,14,19,21,22,24,25,26,27,28,31,33] included details about the underlying disease and/or etiology of heart failure in patients undergoing LVAD, no direct comparison related to the etiology of HF was (i.e., could be) performed, mainly due to wide variation of the survival rates and other outcome measures among patients with different RHF etiologies such as post-cardiotomy or myocarditis. This reflects the fact that the primary cause of RHF and/or indication for t-RVAD were important determinants of outcome following t-RVAD implantation. Although the subgroup analysis was reported in some papers, these studies are obviously not adequately powered to detect a meaningful difference between the different RHF phenotypes/indications for t-RVAD. Furthermore, due to the significant heterogeneity, and the use of different endpoints among included studies, subgroup analyses based on data pooling was not considered in this systematic review. Specific for the underlying disease, almost no outcome was mentioned based on it in any paper.

In brief, the most favorable outcome was found in post-LAVD patients with early planned insertion of pRVAD versus other patients’ phenotypes or delayed insertion or insertion of sRVAD. However, this analysis also is not adequately powered to detect a meaningful difference between the different subgroups.

### 4.2. Duration of t-RVAD Dupport

Impella RP is approved for use as t-RVAD for up to 14 days only [39] also the ProtekDuo Cannulas have not been qualified through in vitro, in vivo, or clinical studies for long-term use (i.e., longer than 30 days) as mentioned by the manufacture [40]. The CentriMag is approved for use as t-RVAD up to 30 days [41]. In some studies [8,13,30], the duration of t-RVAD support exceeded the previous mentioned duration. Vierecke et al. [13] in the EUROMACS registry reported that six months after LVAD + t-RVAD implantation, more than 40% of patients who received t-RVAD remain dependent on the t-RVAD support. However, the authors did not provide details on management of RVADs during that time nor the outcome after that. Development of RHF predictive models to choose between temporary or long-term RVAD and global scientific consensus for the definition of RHF and management algorithms will contribute to solve this issue.

### 4.3. Safety of t-RVAD in This Systematic Review

Definitions of the reported complications following t-RVAD implantation were not clear and standardized between the studies. Acute kidney injury, bleeding, neurological events and device thrombosis were the main reported complications. Furthermore, as these complications related to percutaneously implanted t-RVAD two studies [10,20] reported the incidence of tricuspid and pulmonary valves affection and two studies [10,12] reported the incidence of superior vena cava syndrome.

### 4.4. Limitations

There are several important limitations. First, significant heterogeneity between different studies in definition of reported outcome. Second, small sample size in most of the studies even some studies [19,20,28,30,34] were republished after including more patient to the original cohort published before [42,43,44,45,46]. Third, there were significant differences between the studies in indication, timing, t-RVAD type, approach of t-RVAD insertion, duration of support and follow up periods, making the comparison between the different studies or pooling the data in meta-analysis is not possible.

## 5. Conclusions

Temporary RVAD is a lifesaving option for patients with severe RHF, but the evidence stems from small non-randomized heterogeneous studies utilizing a variety of devices. Both aetiology of RHF and time of intervention might play a major role in determining the temporary RVAD outcome. Standardized endpoints definitions, design and methodology for temporary RVAD trials is needed. Furthermore, efforts should continue to improve the technology as well as improve the timely provision of a temporary RVAD.

## Figures and Tables

**Figure 1 jcm-11-00613-f001:**
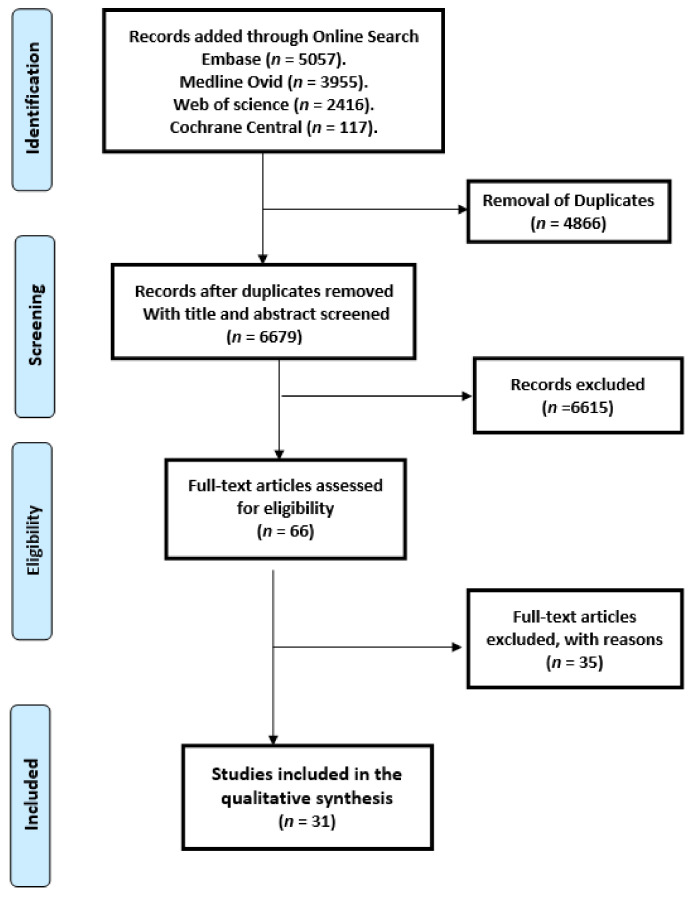
Preferred Reporting Items for Systematic Reviews and Meta-Analyses (PRISMA) flow chart of the data.

**Table 1 jcm-11-00613-t001:** Patients’ characteristics in the 31 studies included in the systematic review.

Author	Indication for t-RVAD; [Underlying Disease for LVAD (%)]/Subgroups (No.)	Patients No.	Male (%)	Age *	Follow Up *
Aissaoui et al., 2014 [30]	Post-LVAD	57	nr	54 ± 14	nr
Anderson et al., 2018 [20]	Post-LVAD (31); PCCS (13); HTX (7); post-RV infarction (9)	60	68	59 ± 15	nr
Badu et al., 2020 [12]	PCCS (18); CM (12); respiratory failure (10)	40	73	55 ± 16	nr
Bhama et al., 2018 [19]	Post-LVAD (42) [ICM (52%), DCM (43%), other (5%)]; PCCS (13); HTX (25)	80	nr	nr	nr
Cheung et al., 2014 [29]	Post-LVAD (2); PCCS (4); myocarditis (2); post-RV infarction (7); HTx (3)	18	67	57 ± 10	365
Coromilas et al., 2019 [17]	Post-LVAD/pRVAD (19); sRVAD (21)	40	85	59 ± 12	nr
Deschka et al., 2016 [25]	Post-LVAD [ICM (56%), DCM (40%), chronic rejection (4%)]	25	80	55 ± 12	575 ± 541
Jaidka et al., 2019 [16]	Pre-valvular surgery	10	40	66 ± 15	nr
Kapur et al., 2013 [34]	Different indications/pRVAD (22); sRVAD (24)	46	nr	nr	nr
Khani-Hanjani et al., 2013 [33]	Post-LVAD [ICM (33%), DCM (67%)]	12	84	51 (24–69)	371
Khorsandi et al., 2019 [15]	Post-LVAD/concurrent t-RVAD (29); staged t-RVAD (14)	43	86	51 (19–76)	453 (2–3560)
Kiernan et al., 2017 [23]	Post-LVAD	386	79	nr	nr
Kremer et al., 2020 [11]	Post-MI	10	90	nr	96 ± 108
Lazar et al., 2013 [32]	Post-LVAD	34	68	52 ± 12	nr
Leidenfrost et al., 2016 [24]	Post-LVAD/t-RVAD only (27) [ICM (80%]; t-RVAD-MO (12)	27	nr	56 ± 15	nr
Loforte et al., 2011 [37]	Post-LVAD (10); PCCS (9)	19	nr	nr	nr
Loforte et al., 2013 [31]	Post-LVAD [ICM (30%), DCM (55%), myocarditis (4%), others (11%)]	46	nr	55 (25–70)	nr
Oliveros et al., 2021 [8]	Different indications, post LVAD (1)	11	54	59 ± 16	nr
Patil et al., 2015 [27]	Post-LVAD [ICM (11%), DCM (80%), myocarditis (3%), CHD (3%), PPCM (3%)]	35	66	40 ± 15	nr
Qureshi et al., 2020 [6]	Different indications	12	67	18	nr
Ravichandran et al., 2018 [18]	Different indication, post LVAD (12)	17	76	56 ± 8	nr
Saeed et al., 2015 [26]	Post-LVAD (17) [ICM (57%), DCM (24%), post MI (19%)]; PCCS (4)	21	71	58 ± 14	nr
Saito et al., 2012 [36]	Post LVAD	26	62	33 ± 15	nr
Salna et al., 2020 [10]	Post LVAD	27	78	63	408
Schaefer et al., 2017 [22]	Post-LVAD [ICM (20%), DCM (50%), myocarditis (10%), post-MI (20%)]; minimally invasive (10); sternotomy (11)	21	10	50 ± 15	274 ± 179
Schmack et al., 2019 [14]	Post-LVAD [ICM (55%), DCM (45%)]	11	91	52 ± 13	215 ± 283
Schopka et al., 2012 [35]	Different indications	12	83	nr	nr
Shekiladze et al., 2020 [9]	Post-LVAD (6); acute PE (9); PCCS (8); post-MI (11); non ischemic CM (5)	39	nr	57 ± 16	nr
Takeda et al., 2014 [28]	Post-LVAD [ICM (41%), DCM (41%), myocarditis (6.8%), others (11%)/weaning group (21); failure group (23)	44	nr	nr	nr
Vierecke et al., 2019 [13]	Post-LVAD [ICM (24%), DCM (26%), myocarditis (6%), CHD (1%)]	342	nr	55 (46–62)	nr
Yoshioka et al., 2017 [21]	Post-LVAD [ICM (44%), DCM (56%)]	27	nr	50 ± 15	nr

* mean ± SD/median (days); Numbers are rounded to the nearest full digit when applicable. Abbreviations: CHD = congenital heart disease; CM = cardiomyopathy; DCM = dilated cardiomyopathy; ICM = ischemic cardiomyopathy; HTx = post-heart transplantation; LVAD = left-ventricular assist device; MI = myocardial infarction; No. = number; nr = not reported; pRVAD = percutaneously implanted temporary right-ventricular assist device (t-RVAD); PCCS = post-cardiotomy cardiogenic shock; sRVAD = surgically implanted t-RVAD; t-RVAD-MO = t-RVAD connected to membrane oxygenator.

**Table 2 jcm-11-00613-t002:** Summary of different types of commonly used t-RVADs.

Device	Features (Description)	Approach for Implantation and Configuration	Duration of Support	Advantages	Disadvantages (Limitations)
**ProtekDuo**	Dual lumen cannula and must connected to extracorporeal pump usually TandemHeart or less frequent CentriMag.	Percutaneously through IJV, inflow in the RA and outflow in the PA.	30 days	Single venous access, IJV access allowing patient to remain ambulatory, oxygenator can be added.	May cause SVC syndrome with larger cannula size.
**Impella RP**	Intra-corporeal dual lumen cannula with microaxial pump with flow rate up to 4.5 L/min.	Percutaneously through Fem. V, inflow in the IVC and the outflow in the PA.	14 days	Single venous access, small dimension of the machine.	No oxygenation capacity, femoral access limit the patient’s mobility.
**CentriMag**	Extracorporeal centrifugal pump up to 10 L/min.	1—Surgically (sternotomy) via direct cannulation of RA (inflow) and PA (outflow). 2—Minimally invasive surgical approach via left-sided mini-thoracotomy to suture the outflow cannula of t-RVAD with PA and t-RAVD inflow via Fem. V. 3—By peripheral cannulation of the Fem. V and direct cannulation of the PA. 4—Percutaneously via ProtekDuo. 5—Percutaneously peripheral cannulation of the Fem. V and a percutaneous PA cannula (Fem. V or IJV).	30 days	Variety of connection methods, oxygenator can be added.	Usually surgically implanted.
**TandemHeart**	Extracorporeal centrifugal pump. flow rate up to 4.5 L/min.	1—Percutaneously via ProtekDuo cannula. 2—Percutaneously peripheral cannulation of the Fem. V and a percutaneous PA cannula (Fem. V or IJV).	30 days	Single venous access, IJV access allowing patient to remain ambulatory, oxygenator can be added.	May cause SVC syndrome with larger cannula size. femoral access limit the patients mobility.

Abbreviations: Fem. V = femoral vein; IJV = internal jugular vein; IVC = inferior vena cava; PA = pulmonary artery; RA = right atrium; SVC = superior vena cava.

**Table 3 jcm-11-00613-t003:** t-RVAD characteristics in the 31 studies included in the systematic review.

Author	Type of t-RVAD Device	Approach of Implantation	t-RVAD Duration *
Aissaoui et al., 2014 [30]	CentriMag (40), Thoratec PVAD (17)	Surgical	32 (3–400)
Anderson et al., 2018 [20]	Impella RP	Fem. V	4 ± 2
Badu et al., 2020 [12]	ProtekDuo(weaned)/(for died)	IJV	14 ± 7/10 ± 12
Bhama et al., 2018 [19]	CentriMag	Surgical	6
Cheung et al., 2014 [29]	Impella RP (3)/Impella RD (15)	Fem. V(RP)/surgical (RD)	7 (2–19)
Coromilas et al., 2019 [17]	ProtekDuo (15)/Impella RP (4)/CentriMag (21)	IJV/Fem. V/surgical	9–18
Deschka et al., 2016 [25]	Biomedicus Bio-Pump or Rotaflow RF32+ Oxygenator	Surgical	11 ± 7
Jaidka et al., 2019 [16]	CentriMag	Surgical	4 ± 1
Kapur et al., 2013 [34]	TandemHeart	Percutaneous (22), surgical (24)	5 ± 5
Khani-Hanjani et al., 2013 [33]	Rotaflow	Surgical	8 (3–18)
Khorsandi et al., 2019 [15]	CentriMag (34), Rotaflow (8), AB5000(1)	Surgical	nr
Kiernan et al., 2017 [23]	nr	nr	nr
Kremer et al., 2020 [11]	ProtekDuo	IJV	10 ± 7
Lazar et al., 2013 [32]	CentriMag	Surgical	nr
Leidenfrost et al., 2016 [24]	CentriMag (25), Impella LD (1), AB5000(1)/+ Oxygenator (12)	Surgical	10 ± 9/5 ± 3
Loforte et al., 2011 [37]	CentriMag (PCCS/post LVAD)	Surgical	(9 ± 3)/(19 ± 9)
Loforte et al., 2013 [31]	CentriMag	Surgical	16 (2–50)
Oliveros et al., 2021 [8]	ProtekDuo	IJV	58 ± 47
Patil et al., 2015 [27]	CentriMag	Surgical	nr
Qureshi et al., 2020 [6]	Impella RP	Fem. V	7 (0.2–18)
Ravichandran et al., 2018 [18]	ProtekDuo	IJV	11 ± 7
Saeed et al., 2015 [26]	CentriMag ± oxygenator (12)	Surgical	9 (2–88)
Saito et al., 2012 [36]	Capiox or Gyropump	Surgical	5 ± 3
Salna et al., 2020 [10]	ProtekDuo	IJV	11
Schaefer et al., 2017 [22]	CentriMag/Deltastream pump	Surgical (minimally invasive)	16 ± 12
Schmack et al., 2019 [14]	ProtekDuo	IJV	17 ± 10
Schopka et al., 2012 [35]	Rotaflow ± oxygenator	Surgical (minimal invasive)	11 (2–43)
Shekiladze et al., 2020 [9]	Impella RP	Fem. V	3
Takeda et al., 2014 [28]	CentriMag (17), AB5000(25), Thoratec PVAD (1)	Surgical	nr
Vierecke et al., 2019 [13]	CentriMag (128), others (214)	NR	nr
Yoshioka et al., 2017 [21]	CentriMag	Surgical	14 (10–18)

* mean ± SD/median (days); Numbers are rounded to the nearest full digit when applicable. Abbreviations: Fem. V = femoral vein; IJV = internal jugular vein; nr = not reported.

**Table 4 jcm-11-00613-t004:** Survival of patients treated with t-RVAD in the 31 studies included in the systematic review.

Author	No.	To Weaning (%)	To Discharge (%)	30 Day (%)	180 Day (%)	Died (%) #
Aissaoui et al., 2014 [30]	57	nr	nr	nr	47	nr
Anderson et al., 2018 [20]	60	nr	nr	72 **	62	27
Badu et al., 2020 [12]	40	73	68	nr	nr	nr
Bhama et al., 2018 [19]	80	nr	nr	64	nr	58
Cheung et al., 2014 [29]	18	78	nr	72	50 *	nr
Coromilas et al., 2019 (pRVAD/sRVAD) [17]	19/21	nr	nr	84/67	nr	nr
Deschka et al., 2016 [25]	25	nr	68	nr	56 *	52
Jaidka et al., 2019 [16]	10	100	100	100	80	20
Kapur et al., 2013 (pRVAD/sRVAD) [34]	22/24	nr	50/38	nr	nr	nr
Khani-Hanjani et al., 2013 [33]	12	nr	92	nr	92 *	8
Khorsandi et al., 2019 (concurrent/staged) [15]	29/14	nr	90/36	93/71	nr	51
Kiernan et al., 2017 [23]	386	nr	nr	78	64	nr
Kremer et al., 2020 [11]	10	40	nr	60	nr	40
Lazar et al., 2013 [32]	34	nr	88	nr	76 *	24
Leidenfrost et al., 2016 (t-RVAD only/t-RVAD-MO) [24]	15/12	nr	nr	53/92	63 ***	nr
Loforte et al., 2011 (PCCS/post LVAD) [37]	9/10	56/80	nr	nr	nr	nr
Loforte et al., 2013 [31]	46	nr	57	74	54	nr
Oliveros et al., 2021 [8]	11	nr	nr	82	72	36
Patil et al., 2015 [27]	35	nr	nr	94	73	nr
Qureshi et al., 2020 [6]	12	nr	83	nr	nr	33
Ravichandran et al., 2018 [18]	17	nr	nr	nr	nr	41
Saeed et al., 2015 [26]	21	nr	62	nr	52 *	38
Saito et al., 2012 (all/weaned) [36]	26/11	nr	nr	nr	nr/82	nr
Salna et al., 2020 [10]	27	nr	nr	nr	81 *	19
Schaefer et al., 2017 (minimally invasive/sternotomy) [22]	10/11	100/nr	nr	80/46	nr	20/nr
Schmack et al., 2019 [14]	11	91	nr	73	nr	36
Schopka et al., 2012 [35]	12	nr	50	nr	nr	nr
Shekiladze et al., 2020 [9]	39	nr	nr	49	nr	nr
Takeda et al., 2014 (weaning group/failure group) [28]	21/23	nr	86/36	nr	75/13	nr
Vierecke et al., 2019 ## [13]	342	nr	nr	73	60	nr
Yoshioka et al., 2017 [21]	27	nr	59	nr	59	41

# Died (%) reported at end of follow up. ## survival rate reported as freedom from death at 30-Days and 180-Days. * The survival % at 1year; ** The survival at 30 days or discharge post-device explant (whichever is longer), or to induction of anesthesia for a long-term therapy; *** both group combined. Numbers are rounded to the nearest full digit when applicable. Abbreviations see Table 1.

**Table 5 jcm-11-00613-t005:** Outcome (other than survival) of patients treated with t-RVAD.

Author	Weaned (%)	Switch to Permanent RVAD (%) (Switch to HTx, %)	ICU Stay (Days) *	AKI/RRT (%)
Aissaoui et al., 2014 [30]	nr	nr—(18)	nr	33
Badu et al., 2020 [12]	73	nr	nr	nr
Bhama et al., 2018 [19]	78	nr	nr	nr
Cheung et al., 2014 [29]	78	nr	nr	nr
Coromilas et al., 2019 (pRVAD) [17]	nr	nr	21 (10–27)	33
Coromilas et al., 2019 (sRVAD) [17]	nr	nr	27 (15–44)	43
Deschka et al., 2016 [25]	92	nr	37 ± 32	36
Jaidka et al., 2019 [16]	100	nr	8	0
Khani-Hanjani et al., 2013 [33]	nr	nr	19 (15–22)	18
Khorsandi et al., 2019 (concurrent) [15]	73	34—(69)	nr	40
Khorsandi et al., 2019 (staged) [15]	71	29—(21)	nr	nr
Kremer et al., 2020 [11]	40	20	16 ± 12	80
Lazar et al., 2013 [32]	92	nr	nr	nr
Leidenfrost et al., 2016 (t-RVAD only/t-RVAD-MO) [24]	66/83	nr	nr	nr
Loforte et al., 2011 (PCCS/post LVAD) [37]	56/80	nr	nr	nr
Loforte et al., 2013 [31]	65	7—(9)	22 (15–50)	15
Oliveros et al., 2021 [8]	55	nr	nr	46
Patil et al., 2015 [27]	nr	nr	23 (5–35)	nr
Qureshi et al., 2020 [6]	75	nr	nr	nr
Ravichandran et al., 2018 [18]	23	35	nr	nr
Saeed et al., 2015 [26]	nr	nr—(10)	nr	52
Saito et al., 2012 [36]	42	nr	nr	nr
Salna et al., 2020 [10]	86	11	36 (22–48)	0
Schaefer et al., 2017 (minimally invasive) [22]	100	nr	nr	40
Schmack et al., 2019 [14]	91	nr	24 ± 17	nr
Schopka et al., 2012 [35]	58	nr	nr	nr
Takeda et al., 2014 [28]	49	nr	nr	nr
Yoshioka et al., 2017 [21]	63	nr	28 (15–35)	41

* mean ± SD/median; Numbers are rounded to the nearest full digit when applicable. Abbreviations: AKI = acute kidney injury; HTx = heart transplantation; ICU = intensive care unit; nr = not reported; PCCS = post cardiotomy cardiogenic shock; pRVAD = percutaneously implanted t-RVAD; RV = right ventricle; RVAD = right ventricular assist device; RRT = renal replacement therapy; sRVAD = surgically implanted t-RVAD; t-RVAD-MO = t-RVAD connected to membrane oxygenator.

**Table 6 jcm-11-00613-t006:** Complications following t-RVAD implantation (%).

Author	Major Hge	GI Hge	Reoperation for Hge	Thrombosis	Stroke	ICH	Sepsis	Pulmonary Hge	Hemolysis
Anderson et al., 2018 [20]	48	nr	nr	nr	nr	nr	nr	0	22
Badu et al., 2020 [12]	0	nr	nr	3	nr	nr	nr	nr	nr
Bhama et al., 2018 [19]	nr	nr	28	nr	nr	nr	55	nr	nr
Cheung et al., 2014 [29]	nr	nr	nr	nr	nr	nr	nr	nr	22
Deschka et al., 2016 [25]	4	12	40	nr	8	8	20	20	nr
Jaidka et al., 2019 [16]	nr	0	0	nr	nr	nr	nr	nr	nr
Kapur et al., 2013 [34]	44	nr	nr	nr	nr	nr	nr	nr	nr
Khani-Hanjani et al., 2013 [33]	nr	nr	36	nr	0	nr	0	nr	nr
Khorsandi et al., 2019 [15]	33	nr	nr	16	23	nr	51	7	nr
Kremer et al., 2020 [11]	40	nr	20	nr	nr	10	nr	nr	nr
Loforte et al., 2013 [31]	43	nr	nr	nr	nr	2	15	9	nr
Oliveros et al., 2021 [8]	nr	46	nr	nr	18	nr	64	nr	nr
Qureshi et al., 2020 [6]	nr	nr	nr	8	nr	nr	nr	nr	42
Ravichandran et al., 2018 [18]	nr	6	nr	nr	nr	12	nr	nr	nr
Saeed et al., 2015 [26]	29	nr	nr	nr	0	nr	19	0	nr
Salna et al., 2020 [10]	nr	nr	nr	4	nr	nr	nr	nr	15
Schaefer et al., 2017 [22]	0	0	nr	0	nr	10	nr	nr	nr
Schmack et al., 2019 [14]	nr	nr	nr	nr	nr	9	nr	nr	nr
Schopka et al., 2012 [35]	0	nr	nr	nr	17	nr	8	nr	nr
Shekiladze et al., 2020 [9]	nr	nr	nr	nr	nr	nr	nr	nr	26
Vierecke et al., 2019 [13]	12	nr	nr	3	3	nr	8	nr	nr
Yoshioka et al., 2017 [21]	nr	nr	nr	nr	19	nr	30	nr	nr

Abbreviations: He = hemorrhage; GI He = gastrointestinal hemorrhage; ICH = intra-cranial hemorrhage; nr = not reported.

## Data Availability

Not applicable.

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
