# Peer review of "Temporary Right-Ventricular Assist Devices: A Systematic Review"

_jcm, 2022, doi:10.3390/jcm11030613_

Round 1
Reviewer 1 Report
Nice review article highlighting the importance of improved studies regarding RVAD's
Author Response
Reviewer #1, comment #1
Nice review article highlighting the importance of improved studies regarding RVAD's.
Response: We thank this reviewer for reviewing our article and for his/her positive comments.
Reviewer 2 Report
I did read with great interest your manuscript entitled “Temporary Right Ventricular Assist Devices: A Systematic Review” reporting a systematic review of temporary mechanical support in acute right heart failure.
After identification and screening of relevant publications, 31 studies were included for analysis and review. Congratulations for the huge work on this very interesting topic.
Indeed the cohorts are very inhomogeneous.
However, there are many data like a summary and there is no sub-group analysis. For example, in most publications, use of a temporary RVAD system was after implantation of a LVAD but there is no information about underlying disease/diagnosis. The underlying disease may be of importance regarding outcome and this information is very important. We need this information.
In addition, there are many questions not answered, e.g. was the surgical approach because the patients underwent open-heart surgery? Was the RVAD implantation in the OR immediately after LVAD implantation?
Finally, regarding devices and approaches, a “simple summary” of techniques maybe is interesting, but we need a sub-group analysis to know when and which technique is in the specific situation justified.
As you mention in the conclusions of your manuscript, these techniques and devices are lifesaving for the very sick patients and therefore we need your analysis to improve outcome.
Reviewer 3 Report
Dear Authors,
Congratulation for your well conducted systematic review regarding the safety and efficacy of temporary right ventricular assist device (t-RVAD) implantation. As you have mentioned, you seem to be the first to assess the outcome of different t-RVADs. The review comprises 31 studies and more than 1500 patients. Thus, the systemic review may constitute a state of the art in the field, even though it lacks the metanalyses because of the heterogeneity of the data.
Some minor observation should be mentioned:
- I consider table 1 redundant, as most of the information are mentioned in the Reference chapter and in table 2. As such, I would advice the authors to remove table 1.
- Did the authors find any variables that influence the efficacity/outcome (surgical procedure, no of days in ICU etc)?
Looking forward to hearing from you,
Round 2
Reviewer 2 Report
Thank you for revision. No other comments.